# Rescue of Hepatic Phospholipid Remodeling Defect in iPLA_2_β-Null Mice Attenuates Obese but Not Non-Obese Fatty Liver

**DOI:** 10.3390/biom10091332

**Published:** 2020-09-17

**Authors:** Walee Chamulitrat, Chutima Jansakun, Huili Li, Gerhard Liebisch

**Affiliations:** 1Department of Internal Medicine IV, University of Heidelberg Hospital, Im Neuenheimer Feld 410, 69120 Heidelberg, Germany; Chutima.Jansakun@med.uni-heidelberg.de (C.J.); Huili.Li@med.uni-heidelberg.de (H.L.); 2Institute of Clinical Chemistry and Laboratory Medicine, University of Regensburg, Franz-Josef-Strauss-Allee 11, 93053 Regensburg, Germany; gerhard.liebisch@klinik.uni-regensburg.de

**Keywords:** PLA2G6, fatty liver, phospholipid remodeling, diet-induced obesity, morbidly obesity, choline and methionine deficiency

## Abstract

Polymorphisms of group VIA calcium-independent phospholipase A2 (iPLA_2_β or PLA2G6) are positively associated with adiposity, blood lipids, and Type-2 diabetes. The ubiquitously expressed iPLA_2_β catalyzes the hydrolysis of phospholipids (PLs) to generate a fatty acid and a lysoPL. We studied the role of iPLA_2_β on PL metabolism in non-alcoholic fatty liver disease (NAFLD). By using global deletion iPLA_2_β-null mice, we investigated three NAFLD mouse models; genetic Ob/Ob and long-term high-fat-diet (HFD) feeding (representing obese NAFLD) as well as feeding with methionine- and choline-deficient (MCD) diet (representing non-obese NAFLD). A decrease of hepatic PLs containing monounsaturated- and polyunsaturated fatty acids and a decrease of the ratio between PLs and cholesterol esters were observed in all three NAFLD models. iPLA_2_β deficiency rescued these decreases in obese, but not in non-obese, NAFLD models. iPLA_2_β deficiency elicited protection against fatty liver and obesity in the order of Ob/Ob › HFD » MCD. Liver inflammation was not protected in HFD NAFLD, and that liver fibrosis was even exaggerated in non-obese MCD model. Thus, the rescue of hepatic PL remodeling defect observed in iPLA_2_β-null mice was critical for the protection against NAFLD and obesity. However, iPLA_2_β deletion in specific cell types such as macrophages may render liver inflammation and fibrosis, independent of steatosis protection.

## 1. Obesity and NAFLD

Obesity is an epidemic with a prevalence rate of 13% of the world’s population [1] and has become a major public health problem resulting in decreased quality of life, reduced working ability, and early death. Obesity-associated co-morbidity and diseases include atherosclerosis, diabetes, non-alcoholic fatty liver disease (NAFLD), and non-alcoholic steatohepatitis (NASH) [2]. A significant proportion of the risk of obesity is due to genetic variance [3,4,5]. Demographics (ethnicity, age, and gender) and behavior (eating behavior, physical activity, and smoking) are environmental factors contributing to obesity as well [6,7,8]. Increased consumption of high-fat-diet (HFD) contributes in a major way to obesity in a genetic variance-dependent manner [7,8]. One example is that C57BL/6J mice are more vulnerable to diet-induced obesity compared to other genetic backgrounds [9,10,11]. The gene-by-diet interactions may be highly heritable and they could significantly have a large impact on obesity in human offspring [12].

A study using more than 100 inbred strains of mice revealed that a high-fat/high-sucrose diet promotes strain-specific changes in obesity that is not accounted for by food intake [13]. This provides evidence for a genetically determined set-point for obesity at least for the case of high-fat/high-sucrose feeding [13]. The genome-wide association studies (GWAS) of obesity data have been used to elucidate the function of genetic variants. The GWAS analyses of knockout mouse phenotypes have provided PLA2G6 association with body weights, lipids, energy, and nervous system [14]. In human studies, the failure to explain a larger fraction of the genetic basis of obesity alone highlights the gene-by-diet interactions, genetic determinants of habitual dietary intake, as well as the interplay between diet, genes, and obesity [13,14].

Hepatic manifestation of obesity is NAFLD [15]. NAFLD is one of the most common causes of chronic liver disease worldwide [16]. NAFLD pathogenesis has a spectrum covering from steatosis through NASH to cirrhosis, which may progress to primary liver cancer [17]. NAFLD prevalence is 27–34% of the general population in the USA, and 40–90% of global obese populations have this disease [16,17]. Similar to obesity, genetic variances [18,19,20], hormones [20], sex [21,22], ethnicity [23] combined with age [24], as well as dietary and physical activity habits [25] are important factors and traits for NAFLD development.

While NAFLD is commonly seen in obese subjects, it is however not rare among non-obese and lean individuals [26,27] particularly those with specific ethnic backgrounds, such as, Asia-Pacific [28,29]. Genetic predispositions, fructose- and cholesterol-rich diet, visceral adiposity, and dyslipidaemia play an important role in the pathogenesis of lean NAFLD [26,27]. Lean-NAFLD patients show less severe histological features as compared to overweight and obese NAFLD patients. For the latter, a significant ~25% increment of mean fibrosis score is found suggesting that obesity could predict a worse long-term prognosis [30]. Lean subjects with evidence of NAFLD have clinically relevant impaired glucose tolerance, low adiponectin concentrations, and a distinct metabolite profile with an increased rate of patatin-like phospholipase containing lipase 3 (PNPLA3) risk allele carriage [31]. Cardiovascular events are the main cause of mortality and morbidity in non-obese NAFLD [26,27]; which is similar to obese NAFLD [32]. As atherogenic dyslipidaemia arises from hepatic steatosis [32], the metabolism of intrahepatic fat in NAFLD is also recognized to contribute to complications of obesity [33]. While lifestyle changes that include physical activity and weight loss are the mainstay of NAFLD treatment, the understanding of hepatic lipid metabolism may provide some clues for specific interactions between nutrients and dietary needs [34]. Thus, the understanding of the balanced biomolecules and nutrients in the diets would become important in providing insights for an alternative strategy to treat and alleviate NAFLD and obesity [35].

## 2. Animal Models of Obese and Non-Obese NAFLD/NASH

In order to identify important biomolecules and nutrients involved in NAFLD, we have used mouse NAFLD models because mice have shorter lifespan and provide research results in a relatively short period of time. We performed our studies using three different mouse models of NAFLD/NASH. They included leptin-deficient Ob/Ob and long-term HFD-fed mice (representing obese NAFLD), and mice fed with a methionine-choline deficient (MCD) diet (representing non-obese NAFLD). All of these mice had C57BL/6 background which is prone for obesity [11]. Ob/Ob and HFD feeding represent over-nutrition NAFLD model with metabolic perturbations, glucose intolerance, and insulin resistance that are common in humans with mild NASH [36]. MCD diet feeding of mice causes no increase in weight and obesity and no insulin resistance thus representing a non-obese NAFLD model with pathological mechanisms that lead to NASH [37].

The induction of NAFLD/NASH by MCD diet is based on an impaired synthesis of phosphatidylcholine (PC) and the subsequent reduced production of very low-density lipoproteins (VLDLs), and this leads to accumulation of hepatic triglycerides (TGs) and hepatic steatosis development [36,37]. Although histological features and inflammatory response of MCD model reflect human NAFLD/NASH, this model does not however resemble human metabolic physiology; in that the levels of serum TGs, cholesterol, insulin, glucose, and leptin are not increased. As MCD-fed mice lose their bodyweight and do not exhibit insulin resistance, these mice may represent a model of non-obese NAFLD/NASH. Interestingly, non-obese NAFLD has also been described in mice deficient with phosphatidylethanolamine *N*-methyltransferase (PEMT), which is the enzyme that converts phosphatidylethanolamine (PE) to PC [38]. This bolsters the notion that altered phospholipid (PL) metabolism and changes in the composition of PC and PE are linked to NAFLD pathogenesis.

In this review, we investigated the extent of hepatic fatty acid (FA) and PL metabolism in livers of male Ob/Ob mice at six months old [39], male C57BL/6 mice at six months old fed with HFD (60 kcal % fat, Research Diet, USA) for six months [40], and female C57BL/6 mice at 12 months old fed with MCD diet (ssniff GmbH, Germany) for four weeks [41].

## 3. Phospholipids in NAFLD/NASH

In NAFLD, hepatic TG contents are the bulk vesicular fat stored in lipid droplets, thus the alteration in TG metabolism has been a focus for NAFLD prevention [42]. However, hepatic PLs could also play a role in NAFLD by three major mechanisms because PLs and their metabolism are important for (1) the formation of lipid droplets [43,44], (2) the regulation of de novo lipogenesis via sterol regulatory element-binding proteins (SREBPs), a family of membrane-bound transcription factors that regulate synthesis of cholesterol and unsaturated FAs [45,46], and (3) the metabolism and secretion of VLDLs [36,37].

In (1), cytosolic lipid droplets are the sites for storage of neutral lipids including TGs, which are surrounded by a monolayer of PLs [43]. It is known that relative abundance of PC and PE on the surface of lipid droplets is important for their dynamics [44]. An inhibition of PC biosynthesis during conditions that promote TG storage increases the size of the lipid droplets [44].

For (2), the disturbance of PC [47] or PE [48] synthesis by respective genetic deletion of *C. elegans* and *Drosophila* leads to an activation of SREBPs. Thus, decreased PL mass due to suppressed synthesis could lead to a compensatory upregulation of SREBPs eventually resulting in an increase in de novo lipogenic lipid synthesis, steatohepatitis, and metabolic syndrome. This notion could be supported by the data using transgenic mice with deletion of PC [47] and PE [48] in the liver; whereby these knockout mice exhibit propensity to develop NAFLD.

For the last (3) case, it has been long known that PLs are required for the formation and stability of lipoproteins [49]. Depletion of PC can affect the endoplasmic reticulum (ER) and protein trafficking in the Golgi [50]. Moreover, a block of the ER-to-Golgi trafficking associated with a decrease in PC synthesis is shown to induce TG accumulation and subsequent lipoprotein secretion [51]. Consistently, ω-3 FA-induced PL remodeling can alter the utilization of TGs in the form of TG-rich lipoproteins [52]. While the ratio of PC/PE that influence hepatocyte membrane integrity can regulate NAFLD [53], hepatocellular PC is shown to exhibit protective effects on hepatic steatosis, however this PC does not protect liver inflammation in NASH [54]. This may indicate a differential role of PC and perhaps other PLs in hepatocytes *versus* in immune cells. Thus, the alteration of hepatic PL metabolism during NAFLD in mice is linked to the syntheses and trafficking of TGs and FAs, as well as the synthesis and secretion of TG-rich lipoproteins.

Accordingly, patients with NAFLD contain a decrease in liver total PC and PE levels, and the contents of arachidonate (20:4)-containing PC and docosahexaenate (22:6)-containing TG are also decreased in livers of NASH patients [55]. The decrease of polyunsaturated fatty acid (PUFA)- containing lipids in NASH livers indicates that there is an impairment of PL remodeling, which could be due to the down-regulation of PL synthesis genes by pro-inflammatory cytokines, such as tumor necrosis factor-α [56]. It is therefore essential to determine PL profiles as a function of unsaturation of not only PC and PE, but also other lipids including sphingomyelin (SM), ceramides (Cer), and cholesteryl esters (CEs). These results will help identify the relative significance of these types of biomolecules and their different roles in obese and non-obese NAFLD.

## 4. Phospholipid-Metabolizing Genes and Phospholipases A2 (PLA_2_) in Obesity and NAFLD

Since hepatocellular phospholipids play important role in NAFLD/NASH, phospholipid- metabolizing genes have thus been inherently subjected to research investigations. These genes may include phospholipase A2 (PLA_2_) such as group IVA PLA_2_ (or cytosolic PLA_2_α), group IIA PLA_2_ (or PLA2G2A or secretory PLA_2_), as well as lipid hydrolases with specificities for diverse substrates such as TGs, PLs, and retinol esters. These lipid hydrolases include PNPLA family consisting of six enzymes, namely PNPLA2 (ATGL or iPLA_2_ξ), PNPLA3 (adiponutrin or iPLA_2_ε), PNPLA4 (iPLA_2_η), PNPLA6 (iPLA_2_δ), PNPLA8 (group VIB iPLA_2_ or iPLA_2_γ), and PNPLA9 (group VIA iPLA_2_ or PLA2G6 or iPLA_2_β) [57]. An ablation of group IVA PLA_2_ [58] or or iPLA_2_γ [59,60] in mice leads to strong and partial protection against diet-induced obesity, respectively. The attenuation of obesity by group IVA PLA_2_ deficiency could likely be due to the reduction of adipocyte differentiation [61], as well as attenuation of neutrophil infiltration and hepatic insulin resistance [62]. BL/6 mice expressing the human PLA2G2A gene when fed with a fat diet showed more insulin sensitivity and glucose tolerance with a mechanism of mitochondrial uncoupling activation in brown adipose tissues [63]. Inhibitor of secretory PLA_2_ reduces obesity-induced inflammation in Beagle dogs [64], and protects diet-induced metabolic syndrome in rats [65]. Inactivation of the group 1B PLA_2_ (PLA2G1B), a gut digestive enzyme, suppresses diet-induced obesity, hyperglycemia, insulin resistance, and hyperlipidemia in C57BL/6 mice [66,67] and attenuates atherosclerosis and metabolic diseases in LDL receptor-deficient mice [68]. Conversely, transgenic mice with pancreatic acinar cell-specific overexpression of the human PLA2G1B gene gain more weight and display elevated insulin resistance when challenged with a high-fat/carbohydrate diet [69]. Moreover, two secreted PLA_2_s, PLA2G5 and PLA2G2E capable of hydrolysis of lipoproteins, are robustly induced in adipocytes of obese mice, and PLA2G5 prevents palmitate-induced M1 macrophage polarization and PLA2G2E moderately facilitates lipid accumulation in adipose tissue and liver [70]. On the contrary to previously mentioned PLA_2_s, mice deficient with ATGL, iPLA_2_ξ, or PNPLA2 when fed with MCD diet show exacerbated hepatic steatosis and inflammation [71]. Interestingly, mice deficient with adiponutrin or PNPLA3 show no protection against HFD or Ob/Ob background [72], but on the other hand show protection under ER stress [73]. Taken together, these publications show a growing list of lipolytic enzymes that act as metabolic coordinators of obesity and NALFD in mice.

Consistent with mouse data, obese human subjects with or without Type-2 diabetes show high activities of total PLA_2_ and of Ca^2+^-dependent and Ca^2+^-independent enzymes; and that Ca^2+^-dependent secretory sPLA_2_ are the main enzyme responsible of obesity-associated high activity [74]. Moreover, sPLA2 activity is increased with high correlation with sensitive C-reactive proteins in morbidly obese patients [75]. Lastly, plasma PLA2 activity is increased in asthma patients and associated with high plasma cholesterol and body mass index [76].

Genome-wide (GWAS) and candidate gene association studies have identified several variants that predispose individuals to developing NAFLD. A study in mouse GWAS has identified 11 genome-wide significant loci to be associated with obesity traits, and a PL-metabolizing enzyme lysophospholipase-like 1 (LYPLAL1) was among these loci identified in the epididymal adipose tissues of diet-induced obese mice [13]. These results are consistent with association of this gene with human NAFLD [77]. When NASH/fibrosis was assessed histologically and non-invasive computed tomography (CT) was used for hepatic steatosis, it is reported that three variants near PNPLA3 are associated with CT hepatic steatosis, and variants in or near LYPLAL1 and adiponutrin or PNPLA3 are associated with histologic lobular inflammation/fibrosis [77]. NAFLD progression has a strong genetic component, and the most robust contributor is PNPLA3 rs738409 encoding the 148M protein sequence variant [78]. Moreover, antisense oligonucleotides-mediated silencing of Pnpla3 reduces liver steatosis in homozygous Pnpla3 148M/M knock-in mutant mice, but not in wild-type littermates fed a steatogenic high-sucrose diet [79]. As the variation in PNPLA3 contributes to ancestry-related differences in hepatic fat content and susceptibility to NAFLD, consistently the weight loss is effective in decreasing liver fat in subjects who are homozygous for the rs738409 PNPLA3 G or C allele [80]. Hence, current data bolster the notion that PL-metabolizing enzymes, particularly PNPLA3, may be involved in NAFLD development and thus may be used as targets for development of drugs for NAFLD/NASH treatment and prevention.

## 5. iPLA_2_β in Obesity and NAFLD and Use of iPLA_2_β-Null Mice

GWAS in >100,000 individuals of primarily European ancestry have identified group VIA calcium-independent PLA_2_ (iPLA_2_β, PLA2G6, or PNPLA9) as one of the 12 loci to be associated with human body fat percentage (BFP) [81]. This is consistent with PLA2G6 association with bodyweight in mice [14]. Further extended GWAS also identified a strong association of PLA2G6 to BFP in metabolically healthy obesity [82]. In this study, the BFP-increasing allele in the locus near PLA2G6 is associated with lower plasma TG levels in men and women, with lower insulin levels and risk of Type-2 diabetes particularly in men, and higher visceral adipose tissue in men [82]. Another GWAS for plasma lipids in >100,000 individuals also identified SNP rs5756931 of PLA2G6 as one of the 95 loci to be associated with plasma TGs [83]. These results were also recently reviewed [84]. While PLA2G6 association with plasma TG is shown to have no effects on cardiovascular disease (CAD) risk [85], PLA2G6 together with PLA2G2 and PLA2G5 levels are however increased in subgroups of patients with CAD [86]. Furthermore, genetic variants at or near PLA2G6 are associated with Type-2 diabetes in European-Americans [87], European-American women [88], and a Chinese population [89]. This is in line with the reported suppressed insulin secretion by islets in response to glucose and forskolin upon global iPLA_2_β deletion in mice [90]. Hence, iPLA_2_β or PLA2G6 may represent a key PL-metabolizing enzyme being critical in the development of obesity and Type-2 diabetes.

iPLA_2_s are lipolytic enzymes not requiring calcium for catalysis in hydrolyzing ester bond of PL at sn-2 position to release a 2-lysoPL and a free FA [91,92,93]. iPLA_2_β is a prototypic iPLA_2_ that is ubiquitously expressed and plays a house-keeping role in PL metabolism and PL remodeling [91,92,93]. iPLA_2_β mediates PL remodeling by regulating the composition of PUFA in PL pools, for example, an increase of PUFA-containing PLs was observed upon treatment of cultured cells with an iPLA_2_β inhibitor [94,95,96].

In 2011, we obtained global-deficient iPLA_2_β-null (KO) mice with exon 9 deletion [90,97] from Dr. John Turk, Washington University School of Medicine, St. Louis, MO, USA. Our first publication in 2016 revealed the functional role of iPLA_2_β inactivation in morbidly obese NAFLD [39]. Male Ob/Ob mice were cross-bred with KO mice. Compared to Ob/Ob mice, the double Ob/Ob-iPLA_2_β KO mice showed protection with significant reduction of body and liver weights, improved glucose tolerance, and reduction in islet hyperplasia [39]. The improvement in hepatic steatosis was also seen by attenuation of liver TG, FA, and CE contents in double Ob/Ob-iPLA_2_β KO mice.

Work from Dr. Turk’s laboratory showed that HFD feeding of iPLA_2_β-null mice for six months did not improve, but rather further impaired glucose intolerance likely due to an impairment of insulin secretion by pancreatic islets [90]. Moreover, the global deletion of exon 2 in the iPLA_2_β gene in mice fed with HFD for eight weeks also did not show any improvement in serum and liver TGs [98]. The lack of effects could be due to relative short HFD feeding such that hepatic PLs were not yet affected. We therefore attempted to define the conditions among the three NAFLD models that an inactivation of iPLA_2_β is effective in alleviating obesity and NAFLD. We were particularly interested in comparing hepatic PL profiles in iPLA_2_β-null mice in obese and non-obese NAFLD. We considered that hepatic PLs and TGs may be affected by the metabolism in adipose tissues of iPLA_2_β-null mice, since iPLA_2_β is shown to regulate adipocyte differentiation [99]. Interestingly unlike PLA2G6, group IVA PLA_2_ or group VIB iPLA_2_ (iPLA_2_γ) are not included in human GWAS data on obesity/adiposity and blood lipids as discussed above [14,81,82,83,84,85,86,87,88,89]. Thus, PLA2G6 or iPLA_2_β may exhibit a unique activity with a preference toward obesity and hence obese NAFLD [39].

We performed HFD feeding of WT and iPLA_2_β KO mice for six months as another model of obese NAFLD [40]. We showed that protection was observed in iPLA_2_β KO mice with an attenuation of HFD-induced body and liver-weight gains, liver enzymes, serum-free FAs, as well as hepatic TGs and steatosis scores. However, this deficiency did not attenuate hepatic ER stress, fibrosis, and inflammation markers. No protection was observed after short-term 3–5 week HFD feeding when hepatic PL contents were not yet depleted.

Since PL syntheses are disturbed by MCD feeding of mice [36,37], we tested whether iPLA_2_β KO mice could still be protected from fatty liver in this non-obese NAFLD model. MCD feeding of female wild-type (WT) for four weeks induced hepatic steatosis with a severe reduction of body and visceral fat weights, which were not altered in MCD-fed iPLA_2_β-KO mice [41]. However, iPLA_2_β deficiency attenuated MCD-induced elevation of serum transaminase activities and hepatic expression of FA translocase Cd36, fatty-acid binding protein-4, peroxisome-proliferator activated receptorγ, and HDL-uptake gene scavenger receptor B type 1 (SR-B1). The reduction of lipid uptake genes was consistent with a decrease of hepatic esterified and un-esterified FAs and CEs [41]. On the contrary, iPLA_2_β deficiency under MCD did not have any effects on inflammasomes and pro-inflammatory markers but rather exacerbated hepatic expression of myofibroblast α-smooth muscle actin and vimentin [41].

Taken together, iPLA_2_β deficiency elicited protection against hepatic steatosis in an order of Ob/Ob › HFD » MCD; or that protection was better in obese NAFLD compared to non-obese NAFLD model.

## 6. Metabolic Lipid Changes in Ob/Ob Mice and Modulation by iPLA_2_β Deficiency

Because the liver does not serve as a storage depot for fat, the steady-state concentration of hepatic TGs is low under physiological conditions. There is nevertheless a considerable trafficking of both TGs and FAs into and out of the liver during NAFLD development induced by genetic alterations, increased fat intake, and/or alteration in hepatic metabolism [34,42]. Hepatic steatosis in NAFLD develops when the rate of FA input is greater than that of FA output. Thus, the mechanisms for gene-to-diet interactions on the extent of steatosis are very complex since many genes are involved in the regulation of TG, FA, and lipoprotein syntheses; and that some of these lipids have been identified as obese [19] and non-obese [100] NAFLD modifier genes.

We first compared hepatic steatosis among WT and Ob/Ob mice. Liver TG and total FA contents were increased in Ob/Ob mice (Figure 1A). Metabolomic profiling has been used to study hepatic lipid metabolism [39,40,41]. Gas chromatography mass spectrometry (GC/MS) method was used to measure un-esterified and esterified FA species present in all lipids [101]. Regarding FA composition (% Mol), Ob/Ob mice showed an increase of hepatic FAs containing monounsaturated FAs (MUFA) but a decrease in those containing di- and >2 unsaturated FAs (Figure 1B).

PC and PE metabolism is important in pathogenesis of fatty liver owing to disrupted membrane integrity and suppressed PC syntheses as well as altered PC and PE composition [102]. Hepatic steatosis is associated with the reduction of hepatic PC or PLs as demonstrated in experiments in transgenic mice with a deletion of a PC synthesis gene [47,48,53]. Here, an electrospray ionization tandem mass spectrometry (ESI-MS/MS) method was utilized to profile PL species including PC, SM, lysoPC (LPC), LPE, PE, phosphatidylserine (PS), phosphatidylinositol (PI), plasmalogens (Pla), Cer, CE, and free cholesterol (FC) [103,104,105,106]. The composition among these PL subclasses (% Mol) showed a decrease of PC, PE, and PI, but an increase of CE in genetic Ob/Ob mice (Figure 1C). This indicates a shift from polar PLs to neutral lipids, namely, TGs and CEs (Figure 1A,C). The increase of CEs in Ob/Ob livers may reflect diabetes and hyperinsulinemia in these mice. Genetically obese Ob/Ob livers showed a significant increase in MUFA-containing CEs concomitant with a decrease of PUFA-containing PC, PE, PS, and PI as well as SM contents (Figure 1D–F).

PL contents and composition in livers of WT, Ob/Ob, and Ob/Ob-iPLA_2_β KO mice were determined by ESI-MS/MS. Liver histology showed marked steatosis attenuation in Ob/Ob-iPLA_2_β KO mice (Figure 2A). Here, PL composition (% Mol) (Figure 2B), MUFA-PL (Figure 2C), and PUFA-PL (Figure 2D) contents were analyzed showing the suppression of PUFA-PC, PUFA-PE, and PUFA-PS contents in Ob/Ob livers. This suppression was reversed in Ob/Ob-iPLA_2_β KO mice. Moreover, the elevation of MUFA-CEs and PUFA-CEs in Ob/Ob livers was also attenuated in Ob/Ob-iPLA_2_β KO mice (Figure 2C,D). These changes were associated with an attenuation of bodyweight gains, hepatic steatosis, as well as the reduction of hepatic and plasma TGs [39]. These results showed that iPLA_2_β has a pathophysiological function by depleting PUFA concentrations in Ob/Ob liver PLs. iPLA_2_β inactivation re-establishes PL remodeling to return to normal homeostasis.

## 7. Metabolic Lipid Changes in HFD-Fed Mice and Modulation by iPLA_2_β Deficiency

Livers of WT mice fed with HFD showed a significant increase of TGs and FAs (Figure 3A) [40]. On the contrary to Ob/Ob mice (Figure 1B), hepatic FA composition showed an increase of di-unsaturated FAs by HFD feeding (Figure 3B). For HFD-fed mice, PL composition plot (% Mol) showed a weaker shift from PC to CEs (Figure 3C) when compared with Ob/Ob mice (Figure 1C). There was a decrease in liver MUFA-PC (Figure 3D) concomitant with an increase in SM (Figure 3F). Hence, hepatic PC particularly PUFA-PC was the key PL that was suppressed in HFD obese NAFLD model. This suppression indicated a defect in hepatic fatty-acyl PC remodeling. It appears that PUFA-PE contents were modulated differently between Ob/Ob (Figure 1E) and HFD (Figure 3E) obese NAFLD.

PL contents and composition in livers of WT, HFD-fed WT, and HFD-fed iPLA_2_β KO mice were determined by ESI-MS/MS. Liver histology showed marked steatosis attenuation in HFD-fed iPLA_2_β KO mice (Figure 4A). Analyses of hepatic PL composition showed that the elevation of liver CEs in HFD-fed WT mice was attenuated by iPLA_2_β deficiency (Figure 4B). In HFD model, iPLA_2_β deficiency was rescued by loss of PUFA-PC and PUFA-PE (Figure 4D) and only a trend rescue of MUFA-PE and MUFA-PS (Figure 4C). Hence, similar to Ob/Ob mice (Figure 2D), the loss of PUFA-PC and PUFA-PE was rescued by iPLA_2_β deficiency in HFD obese model (Figure 4D).

## 8. Metabolic Lipid Changes in MCD-Fed Mice and Modulation by iPLA_2_β Deficiency

Unlike Ob/Ob and HFD obese models, hepatic steatosis was not protected by iPLA_2_β deficiency in non-obese mice fed with MCD diet [41] (Figure 5A). There was a decrease in composition of liver PC but an increase of CEs by MCD feeding (Figure 5B). iPLA_2_β deficiency attenuated the elevation of CEs in composition plot (Figure 5B) and CE contents (Figure 5C) associated with attenuation of SR-B1 by iPLA_2_β deficiency [41]. MCD feeding of WT mice decreased the contents of MUFA-PC, MUFA-PE (Figure 5C), and PUFA-PC (Figure 5D) concomitant with a significant increase in total Cer (Figure 5E), and all these changes were not altered by iPLA_2_β deficiency.

## 9. PL in Liver Endoplasmic Reticulum of HFD- or MCD-Fed Mice and Modulation by iPLA_2_β Deficiency

Because iPLA_2_β is localized in the ER [107] where PL syntheses take place [102], we surmise that PLs in the ER membrane during obese and non-obese NAFLD could be modulated by iPLA_2_β inactivation. In support of this notion, NAFLD induced by HFD feeding of PEMT-knockout mice [108] and genetic obese Ob/Ob mice [109] are associated with changes of PC and PE in liver ER fractions. We determined whether PL contents in the ER could be affected by HFD [40] or MCD diet [110] feeding and in combination with iPLA_2_β deficiency. Our ER preparations from livers led to an enrichment of a resident ER protein calnexin in ER fractions (but not in liver homogenates) [110]; thus confirming the purity of ER membranes for lipidomic measurements.

PL profiles of liver ER fractions of HFD-fed mice were analyzed as PL subclasses (Figure 6A). HFD feeding of WT mice depleted ER PC contents and iPLA_2_β deficiency showed a rescue trend. A similar pattern of a rescue-trend effect of iPLA_2_β deficiency could be observed for ER PE and ER PS. Due to substrate depletion of PC synthesis [37], MCD feeding of WT mice caused a strong reduction of ER PC and ER PE (Figure 6B) [110]. iPLA_2_β deficiency under MCD further suppressed ER PE contents, particularly, those containing PUFA. This deficiency did not, however, have any effects on ER PC contents suggesting specificity iPLA_2_β towards PE in the ER. Hence, MCD-induced defect of ER PL remodeling became more severe by iPLA_2_β deficiency [110].

## 10. Hepatic PL Ratio among Obese and Non-Obese NAFLD and Modulation by iPLA_2_β Deficiency

The importance of maintaining an appropriate hepatic PC/PE ratio has been extensively studied by D. Vance’s research group using PEMT-knockout mice [53,54,102,108]. The clinical relevance of this ratio has been shown that the proportion of patients with NASH have a lower hepatic PC/PE ratio compared to healthy subjects [53]. Interestingly, both low and high hepatic PC/PE ratios in different NAFLD models are associated with an increase in NAFLD scores [102]. The lower PC/PE ratio seen with a deficiency of PEMT, betaine:homocysteine methyltransferase, or CPT:phosphocholine cytidyltransferaseα, correlates with increased NAFLD severity [102]. In contrast, mice with the deficiency of glycine *N*-methyltransferase [102], Ob/Ob mice [109], and mice fed with high fat/high cholesterol/cholate diet [111] show a higher hepatic PC/PE ratio. Hence, hepatic changes in PL composition and PC/PE ratio may be dependent on the experimental models distinguishing between genetic *versus* diet or obese *versus* non-obese NAFLD.

To this end, we investigated whether the ratios among PL subclasses and CEs would indicate hepatic steatosis among our three NAFLD mouse models. The ratios were calculated and separated into groups with the indicated PL subclasses used in the calculation including total PLs, saturated (sat) PLs, MUFA-PLs, and PUFA-PLs (Figure 7). Consistent with previous report in Ob/Ob mice [109], PC/PE ratio among total PLs in our Ob/Ob livers was increased from 1.5 to 2.3; and this increase was seen among sat and PUFA-PLs (Figure 7A). With a significant increase in CEs in Ob/Ob livers (Figure 1D), PC/CE and PE/CE ratios were therefore decreased with genetic obesity (Figure 7A). For HFD-fed WT mice, PC/PE was decreased among total and sat PLs, but a marked decrease was observed in PC/CE and PE/CE among MUFA-PLs and PUFA-PLs (Figure 7B). For MCD-fed WT mice, PC/PE was decreased from 1.5 to 0.5 among total and PUFA-PLs (Figure 7C). MCD feeding caused a marked decrease in PC/PS and PC/PI seen in MUFA-PLs. With a significant decrease in CEs in MCD-fed livers (Figure 5C), PC/CE and PE/CE ratios were therefore decreased in sat- and PUFA PLs (Figure 7C).

Our data showed PC/PE among total PLs increased in Ob/Ob, but on the other hand, decreased in MCD livers. Thus, PC/PE ratios are changed in a U-shape curve from genetic obesity to non-obese NAFLD/NASH [102]. In addition to PC/PE, ratios among other PLs were also altered with fatty liver. A weak decrease in PE/PI was observed in Ob/Ob livers (Figure 7A). A decrease in PE/PI was observed in HFD-fed WT mice (Figure 7B). A weak decrease of PC/PS and PC/PI ratios was observed in MCD-induced NAFLD (Figure 7C). Such changes during NAFLD can support the changes in electrostatics of PL bilayers and the existence of asymmetric lipid membranes due to charged anionic PLs, such as PS and PE relative to PC [112]. This may correlate with the least extent of hepatic inflammatory status in Ob/Ob mice as compared with HFD- and MCD-fed mice [39,40,41].

iPLA_2_β deficiency did not alter PC/PE ratio in Ob/Ob (Figure 7A) and MCD-fed mice (Figure 7C). This deficiency however reversed the suppression of PC/PE and PE/PI in HFD-fed mice (Figure 7B). Since marked exacerbation of cholesterol metabolism is reported in Ob/Ob [113] and diabetic mice [114], the elevation of CEs caused a decrease of PC/CE and PE/CE in all three NAFLD models. iPLA_2_β deficiency reversed the suppression of PC/CE and PE/CE ratios in Ob/Ob and HFD-fed mice, but not in MCD-fed mice (Figure 7A–C). This may suggest that the remodeling with a shuttling of PUFA and MUFA could occur between PC, PE, and CEs via acylation and transacylation in choline/methionine-rich Ob/Ob and HFD livers [115]. Taken together, we have demonstrated a difference between genetic and diet (HFD and MCD) NAFLD regarding the ratios among PL subclasses, a contribution of PS and PI relative to PC and PE, as well as CE metabolism.

## 11. iPLA_2_β and De Novo Lipogenesis Gene Expression in Livers of Mice in 3 NAFLD Models

Associated with hepatic steatosis protection in obese but in non-obese NAFLD models [39,40,41], we further compared expression of iPLA_2_β protein and de novo lipogenesis in livers of Ob/Ob, HFD- and MCD-fed WT mice. Rather than an increase, a slight decrease in iPLA_2_β protein expression was observed in fatty livers of obese models (Figure 8A,B) and a strong decrease in MCD model (Figure 8C). iPLA_2_β mRNA expression was not markedly altered (not shown) [39,40,41]. These data imply that iPLA_2_β protein may be subjected to degradation at post translational levels during NASH. It is shown that iPLA_2_ expression is decreased in rat cirrhotic livers [116], and may support iPLA_2_β as a target for degradation during severe liver injury. Currently, no published data on iPLA_2_β or PLA2G6 expression in livers of NAFLD/NASH patients are available. We could not correlate iPLA_2_β protein expression observed in our results with human data.

With obesity and fatty liver, livers of Ob/Ob and HFD-fed mice showed marked elevation of de novo lipogenesis genes including fatty acid synthase (FAS) and transcription factor SREBP1c (Figure 8A,B). It was shown that the transcription of iPLA_2_β is regulated by SREBP-1 [117], consistently attenuated expression of FAS and SREBP1c was observed in livers of iPLA_2_β-deficient obese mice. This attenuation was in indeed correlated with hepatic steatosis protection. On the other hand, MCD feeding of WT mice caused suppressed expression of these genes, which was not altered by iPLA_2_β deficiency (Figure 8C). This was associated with no steatosis protection by iPLA_2_β deficiency in this non-obese model.

## 12. Summarized PL Characteristics in Ob/Ob, HFD-, and MCD-Fed Mice and Effects of iPLA_2_β Deficiency

Because PC and PE are the two major zwitterionic PLs in cells and their metabolism has been a focus as a key mechanistic base for healthy liver, the alterations in PC and PE are critical in liver disease development and NAFLD [53,54,102,108]. iPLA_2_β deficiency protects obesity and NAFLD with an order of - Ob/Ob [39] › HFD [40] » MCD [41]; and the latter showed no steatosis protection (Figure 9).

Human NAFLD and NASH are associated with numerous changes in the lipid composition of the liver. A decrease of the total PC and a decrease of arachidonic acid (20:4n-6) in FFA, TGs, and PC are reported in both NAFLD and NASH [55]. The contents of eicosapentanoic acid (20:5n-3) and docosahexanoic acid (22:6n-3) are decreased in NASH livers. In another study, PUFA-PLs are decreased in NASH livers compared to normal livers, and liver CEs are increased in NAFLD and NASH livers compared to normal livers [118]. Thus, a defect in PL remodeling and increased CEs could be observed in livers of human NAFLD and NASH. It is reported that the activity of the desaturase FADS1 is decreased in NAFLD liver biopsies [119]. This decrease in desaturation of FFA would likely lead to a depletion of MUFA and PUFA lipids, particularly PLs in NAFLD/NASH. It is shown that the hepatic PC/PE ratio is decreased in human NASH livers [53]. In another study, this ratio is lower in simple steatosis and NASH patients compared with controls, but it was not different between SS and NASH [120]. PC was lower and PE higher in the liver of simple patients compared with controls, whereas in NASH patients, only PE was higher [120]. Thus, the decrease of hepatic PC/PE ratio is a key parameter for human NAFLD and progression to NASH.

Livers of C57b/S129J mice fed a high-fat/high-cholesterol diet show an increase of hepatic CEs while hepatic PC, PE, and PS contents are decreased in NAFLD and a further decrease in PC and PE are observed in NASH [121]. Not only MUFA-PLs, hepatic PUFA-PC, PUFA-PE and PUFA-PI are reported to be decreased in MCD-fed mice compared with chow-fed or HFD-fed mice [122]. Hepatic 16:0 CEs are also increased in MCD-fed mice compared to chow-fed mice.

Results on liver lipids in our studies overall are consistent with those reported in human [55,118,119,120] and mouse [121,122] NAFLD/NASH. In our studies, a decrease of PUFA-PLs and MUFA-PLs was observed in livers of Ob/Ob and HFD-fed mice (Figure 9A) and MCD-fed mice (Figure 9B). The elevation of CEs led to a decrease of PC/CE and PE/CE ratios in obese and non-obese lives. iPLA_2_β deficiency rescued not only the defect of liver PL remodeling, but also reversed the suppression of PC/CE and PE/CE in Ob/Ob and HFD-fed mice (Figure 9A). iPLA_2_β deficiency in MCD-fed mice did not alter these parameters (Figure 9B). In our models, iPLA_2_β deficiency did not interfere with liver sphingolipids. Our work suggests the functions of iPLA_2_β on the hepatocyte PL remodeling in obese NAFLD models. Interestingly, hepatic PC/PE ratio is increased in Ob/Ob mice but decreased in HFD- and MCD-fed mice (Figure 9). Thus, this ratio is a marker in discriminating genetic *versus* diet-induced NASH, which is in a similar manner to human NASH [53,120]. iPLA_2_β deficiency reversed PC/PE ratio in obese but not non-obese model, rendering this ratio as a marker for phenotypic changes in NASH.

Besides PC and PE, our current data present an additional evidence for changes in other PLs including anionic PS and PI in Ob/Ob mice (Figure 9A). iPLA_2_β deficiency elicited full protection against fatty liver, obesity, and elevation of liver enzymes in Ob/Ob and HFD-fed mice [39,40]. In these obese models, we propose that iPLA_2_β specifically hydrolyzes PUFA-PLs and MUFA-PLs for generation of FAs subsequently utilized for TG and CE syntheses for hepatic steatosis. This process may be coordinated with other PLA_2_ enzymes. By this way, PUFA-PL and MUFA-PL contents in Ob/Ob and HFD livers are suppressed, and this suppression is rescued or replenished by iPLA_2_β deficiency. In line with this, administration of n-3 essential FAs [123] and PUFAs [124] have been shown to ameliorate hepatic steatosis in obese mice likely by increasing membrane fluidity [125]. Unlike Ob/Ob mice [39], iPLA_2_β deficiency during HFD feeding did not protect mice from liver inflammation [40]. Consistently, PLs, such as PC, are shown to elicit protection of hepatic steatosis in NAFLD without attenuating liver inflammation in NASH [54]. Alternatively, iPLA_2_β deletion by birth under the background of leptin deficiency may render a complete protection [39]; possibly due to adaptation in different cell types upon gene deletions throughout the mouse lifetime. Chronic HFD feeding on the other hand would represent an external stress to mice [40]. As iPLA_2_β KO mice were global deletion, we therefore proposed that iPLA_2_β deletion may affect specific cell types, such as hepatocytes, immune cells, and adipocytes, differently in response to HFD feeding (Figure 9A).

Due to the lack of choline and methionine in the diet, MCD feeding limits PC synthesis thus resulting in a defect of PL remodeling (Figure 9B). iPLA_2_β deficiency did not protect mice from MCD-induced fatty liver, but attenuated elevation of liver enzymes (Figure 9B). This attenuation could be due to an inhibition of the uptake of FA as determined by Cd36 expression and FA contents, as well as an inhibition of HDL reverse transport as determined by SR-B1 expression and CE contents [41]. The latter may indicate an involvement of iPLA_2_β in cholesterol esterification to CEs [114,115]. Despite attenuation of liver enzymes, iPLA_2_β inactivation during MCD however showed an increase of α-smooth muscle actin and vimentin expression [41]. Such increased liver fibrosis could be due to iPLA_2_β inactivation in specific cell types such as macrophages or hepatocytes associated with stress induced by MCD feeding (Figure 9B).

Taken together, our results highlight the significance of cross-talk between the metabolism of PLs and neutral lipids, i.e., CEs [115,126] and TGs in lipid droplets [43,127], which can be modulated by iPLA_2_β deficiency. In the latter case, iPLA_2_β may co-function with other PLA_2_s such as group IVA PLA2 in the shuttling of FAs toward TG synthesis [127]. iPLA_2_β inactivation was effective in attenuating obese (Ob/Ob and chronic-HFD) NAFLD indicating specific involvement of iPLA_2_β in obesity pathogenesis. As iPLA_2_β inactivation was ineffective to treat non-obese MCD NAFLD, this indicates that choline and methionine in hepatic PC and PL synthesis and metabolism were necessary for protection in obese models. Our data also emphasize the contributions and involvement of hepatic PL, TG, and CE metabolism in the development of NAFLD.

## 13. Perspectives

### 13.1. Consideration of Cell-Type Specificity of iPLA_2_β

The phenotypes of iPLA_2_β-null mice have been recently reviewed [92,93]. On one hand, iPLA_2_β is detrimental in mediating ER stress and cell death of pancreatic β-cells. On the other hand, iPLA_2_β plays a homeostatic role and the loss of iPLA_2_β in mice leads to ageing-related diseases, such as male infertility, bone-density loss, and neurological disorders. Accordingly, PLA2G6 mutations lead to the pathogenesis of infantile neuroaxonal dystrophy and PARK14-linked Parkinson’s disease [92,93]. These data have highlighted various and often opposing functions of iPLA_2_β in a cell-type specific manner.

In our investigations of obese and non-obese NAFLD/NASH, global deletion iPLA_2_β-null mice were used. Thus, we could not identify whether the observed effects of iPLA_2_β inactivation were due to altered functions in adipocytes [99], immune cells such as macrophages [128,129], and Kupffer cells [130], as well as hepatocytes as shown by our work [39,40,41]. Concurrently, we have also reported that global deletion of iPLA_2_β was able to sensitize hepatocellular damage induced by concanavalin A [131] or during ageing [132]. It is thought that such epithelial damage caused by iPLA_2_β deficiency may secrete mediators that in turn activate inflammatory macrophages leading to sensitized injury. The observed sensitization of liver injury [131,132] could therefore be the combined effects of iPLA_2_β deficiency in hepatocytes and macrophages.

The effects of iPLA_2_β inactivation in macrophages have been reported [128,129]. On one hand, macrophages from iPLA_2_β-null mice exhibited suppressed pro-inflammatory M1 response [128,130] and showed enhanced IL-4-induced M2 polarization in vitro [128]. Consistently, iPLA_2_β-null mice treated with anti-CD95 antibody primed Kupffer cells for attenuated release of TNF-α but enhanced release of interleukin-6 in vitro [130]. iPLA_2_β-null mice showed the inability to phagocytose infected parasites in vivo [133]; thus iPLA_2_β deficiency could lead to a defect in innate immunity. As iPLA_2_β-null mice showed propensity for increased liver inflammation and fibrosis during HFD and MCD NAFLD (Figure 9), this could be due to the ability of iPLA_2_β-deficient macrophages to differentially regulate M1 and M2 cytokines during NAFLD.

Our work has demonstrated iPLA_2_β activity in the hepatocytes by a decrease of products lysoPL and an accumulation of substrates PC and PE observed in livers of iPLA_2_β-null mice fed with chow [39,40]. Such accumulation could therefore lead to replenishment of PLs and steatosis protection in obese Ob/Ob and HFD NAFLD (Figure 9A). On the other hand, the alterations of hepatocellular PL composition and a decrease of PC/PE in chow-fed iPLA_2_β-null mice may render altered PL membranes leading to susceptibility for previously observed liver injury [131,132]. Further studies are warranted to determine whether iPLA_2_β inactivation specifically in hepatocytes could affect hepatic steatosis and inflammation/fibrosis in HFD (Figure 9A) and MCD (Figure 9B) models.

While knockdown of iPLA_2_β inhibits hormone-induced differentiation of adipocytes in vitro [99], iPLA_2_β inactivation in adipocytes may inhibit adipocyte expansion, thus attenuating adipogenesis and obesity observed in our obese NAFLD models [39,40]. It has recently been shown that protection of hepatic steatosis by an ablation of adipocyte PLA_2_ is mediated by adipocyte hormone leptin [134]. By unknown mechanisms, iPLA_2_β inactivation may lead to an increase secretion of leptin, and may thus elicit protection in a similar way as adipocyte PLA_2_. Accordingly, leptin has biological activity in depleting liver TG [135] as well as activating Kupffer cells and thereby altering hepatic lipid metabolism [136]. To figure out the contribution of adipocytes, macrophages, and hepatocytes on the NAFLD/NASH pathogenesis, the generation of tissue-specific iPLA_2_β-deficient mice is therefore warranted. These results will help us understand that iPLA_2_β deficiency in which cell type is responsible for steatosis protection in HFD NAFLD and for increased liver fibrosis in MCD NAFLD. Results from tissue-specific iPLA_2_β KO mice will also help with the designs of iPLA_2_β inhibitors and formulations for specific-tissue delivery for effective treatment of obese and non-obese NAFLD.

### 13.2. Use of PLs or iPLA_2_β Antagonists for Steatosis Protection in Obese Versus Non-Obese NAFLD

Because there are no approved drugs for treatment of NAFLD/NASH, our research results may provide some insights for further development toward NAFLD treatment. Here, our data support the idea for repletion of PL loss by use of PLs themselves or use of iPLA_2_β inhibitors for treatment of obese NAFLD. Some iPLA_2_β inhibitors have been found to be effective for treatment of diabetes at least in experimental animals [137]. Potent and selective inhibitors of iPLA_2_β have been developed [138]. Pending the results on the phenotypes of hepatocyte-, macrophage-, and adipose-specific iPLA_2_β KO mice in obese NAFLD models, investigators may design iPLA_2_β inhibitors in combination with specific delivery to hepatocytes [139], macrophages [140], or adipocytes [141] for better treatment of the common disease obese NAFLD.

For non-obese MCD NAFLD, iPLA_2_β inhibitors may be found to be partially effective in attenuating liver enzymes but not hepatic steatosis (Figure 9B). Choline supplementation in patients under parenteral nutrition reverses hepatic steatosis [142], and the strategy for non-obese NAFLD treatment may involve the use of choline and methyl donors as co-treatment with iPLA_2_β inhibitors. Choline deficiency in humans has been shown to induce hepatic steatosis [143,144] causing liver dysfunctions [145]. Hence, supplementation of choline [142] and methyl donors [146] may be effective in attenuating hepatic steatosis under non-obese NAFLD/NASH. Further investigations in experimental animals are necessary to evaluate the long-term use of iPLA_2_β inhibitors alone for obese NAFLD and in combination with choline/methyl donor supplementation for non-obese NAFLD models. Nonetheless, the hierarchical mode of action by iPLA_2_β deficiency indicates that an iPLA_2_β inhibitor may be designed perhaps with tissue-specific delivery for therapeutic development to treat metabolic syndromes due to obese NAFLD, and may not be suitable for non-obese NASH.

## 14. Conclusions

We demonstrated a pivotal role of iPLA_2_β in the development of hepatic steatosis and inflammation in obese and non-obese NAFLD models. iPLA_2_β inactivation rescued the defect in PL remodeling and elicited steatosis protection in obese NAFLD models, but not in non-obese MCD model. While our study suggests the use of iPLA_2_β inhibitors for therapy of obese NAFLD due to genetics or chronic HFD intake, further investigations using tissue-specific iPLA_2_β-deficient mice are still warranted. Here, the usefulness of the lipidomics methodology is shown in deciphering the alterations in hepatic PL pools and ratios among three NAFLD models.

## Figures and Tables

**Figure 1 biomolecules-10-01332-f001:**
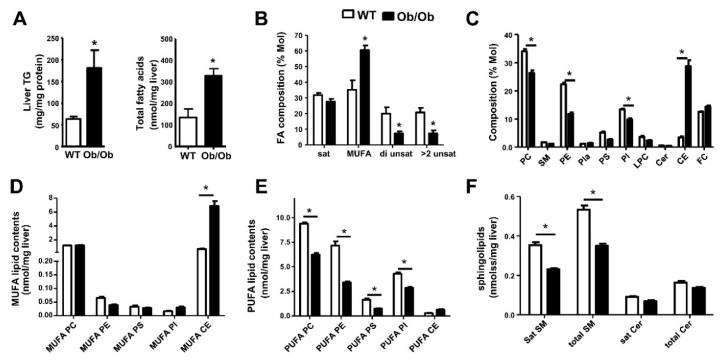
Lipid contents and composition show a defect of hepatic PL remodeling in Ob/Ob mice. Male wild-type (WT) and Ob/Ob mice at six months old were used. (**A**) The contents of hepatic triglycerides (TG), total fatty acids (FAs). (**B**) FA composition of saturated, monounsaturated fatty acids (MUFA), di unsaturated, and > 2 unsaturated FA. (**C**) The composition of phospholipids (PLs) and cholesterol esters (CEs). (**D**) The contents of PLs and CEs containing monounsaturated fatty acids (MUFA). (**E**) The contents of PLs and CEs containing polyunsaturated fatty acids (PUFA). (**F**) The contents of sphingolipids. Data are mean ± SEM, N = 5–7; *, *p* < 0.05 *versus* WT.

**Figure 2 biomolecules-10-01332-f002:**
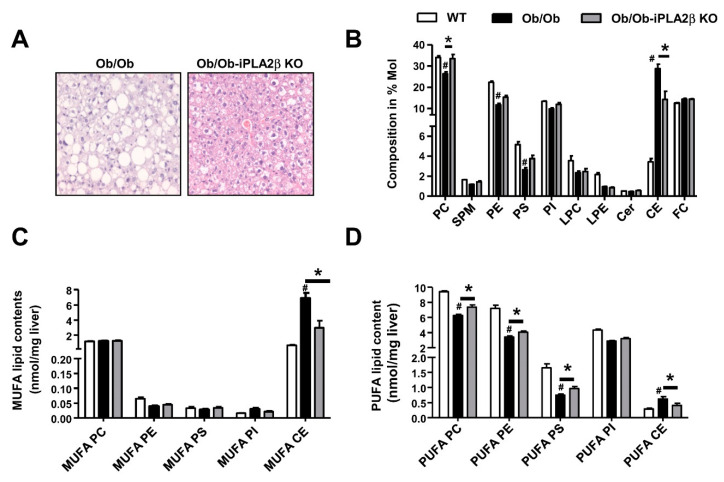
iPLA_2_β inactivation in Ob/Ob mice rescues the defect of hepatic PL remodeling. Male WT, Ob/Ob, and double knockout Ob/Ob-iPLA_2_β KO at six months old were used. (**A**) Representative photographs of hematoxylin- and eosin-stained livers of Ob/Ob mice and Ob/Ob-iPLA_2_β KO. (**B**) The composition of PLs and CEs. (**C**) The contents of PLs and CEs containing MUFA. (**D**) The contents of PLs and CEs containing PUFA. Data are mean ± SEM, N = 5–7; #, *p* < 0.05, *versus* WT; *, *p* < 0.05, Ob/Ob *versus* Ob/Ob-iPLA_2_β KO.

**Figure 3 biomolecules-10-01332-f003:**
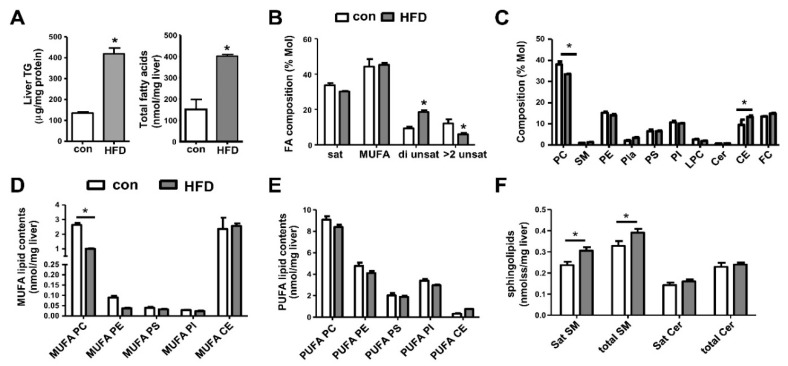
Lipid contents and composition show a defect of hepatic PL remodeling of WT mice fed with high-fat diet (HFD) for six months. Male WT mice at six months old were used. (**A**) The contents of hepatic triglycerides (TG), total fatty acids (FA). (**B**) FA composition of saturated, MUFA, di unsaturated, and > 2 unsaturated FA. (**C**) The composition of PLs and CEs. (**D**) The contents of PLs and CEs containing MUFA. (**E**) The contents (nmol/mg liver) of PLs and CEs containing PUFA. (**F**) The contents of sphingolipids SMs and Cers. Data are mean ± SEM, N = 5–12; *, *p* < 0.05, *versus* WT.

**Figure 4 biomolecules-10-01332-f004:**
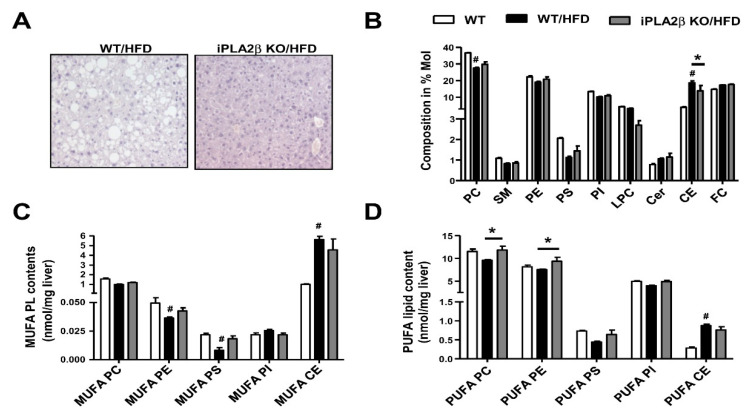
Inactivation in HFD-fed mice rescues the defect of hepatic PL remodeling. Male WT and iPLA_2_β KO mice at six months old were fed with HFD for six months. (**A**) Representative photographs of hematoxylin- and eosin-stained livers of HFD-fed WT and iPLA_2_β KO mice. (**B**) The composition of PLs and CEs. (**C**) The contents of PLs and CEs containing MUFA. (**D**) The contents of PLs and CEs containing PUFA. Data are mean ± SEM, N = 5–12; #, *p* < 0.05, *versus* WT; *, *p* < 0.05, WT/HFD *versus* iPLA_2_β KO/HFD.

**Figure 5 biomolecules-10-01332-f005:**
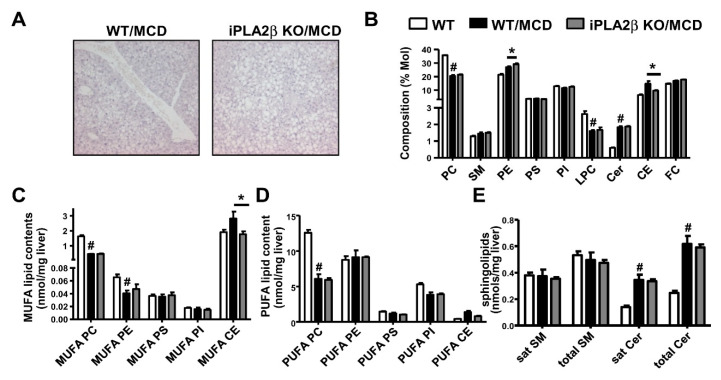
Inactivation does not rescue the defect of hepatic phospholipid remodeling in methionine- and choline-deficient (MCD) diet-fed mice. Female WT and iPLA_2_β KO mice at 12 months old were fed with MCD diet for four weeks. (**A**) Representative photographs of hematoxylin- and eosin-stained livers of MCD-fed WT and iPLA_2_β KO mice. (**B**) The composition of hepatic PLs, CEs, and free cholesterol (FC). (**C**) The contents of PLs and CEs containing MUFA. (**D**) The contents of PLs and CEs containing PUFA. (**E**) The contents of sphingolipids SMs and Cers. Data are mean ± SEM, N = 5–6; #, *p* < 0.05, *versus* WT; *, *p* < 0.05, WT/MCD *versus* iPLA_2_β KO/MCD.

**Figure 6 biomolecules-10-01332-f006:**
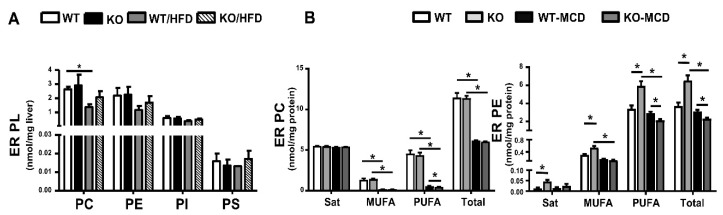
iPLA_2_β inactivation on PL profiles of liver ER fractions of HFD- and MCD-fed mice. Feeding with HFD or MCD diet was described in Figure 4 and Figure 5, respectively. ER fractions were isolated from mouse livers and ER proteins subjected to PL profiling by LC-MS/MS. (**A**) The contents of PC, PE, PI, and PS in the ER of WT, iPLA_2_β-KO, WT/HFD, and iPLA_2_β-KO/HFD livers. (**B**) Saturated, MUFA, PUFA, and total contents of PC and PE in liver ER fractions of WT, iPLA_2_β-KO, WT/MCD, and iPLA_2_β-KO/MCD livers. Data are mean ± SEM, N = 5–12 for (A) and 5–6 for (B); *, *p* < 0.05, between indicated pairs.

**Figure 7 biomolecules-10-01332-f007:**
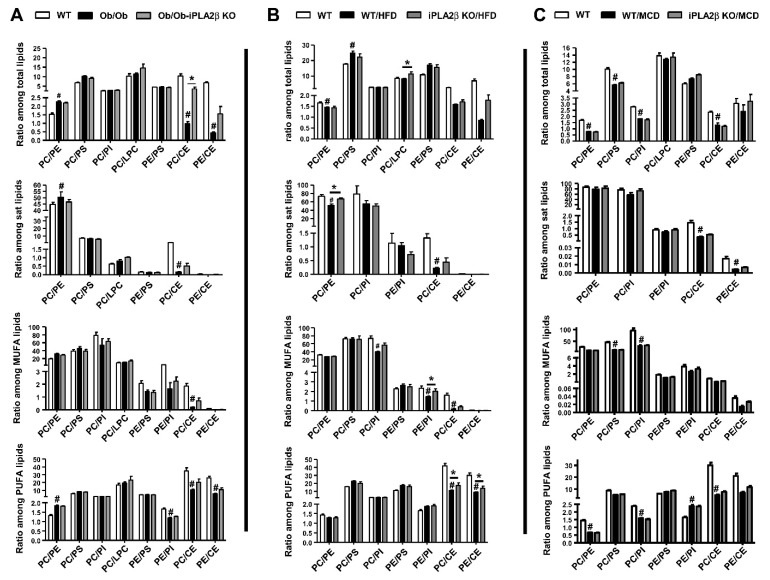
Alters the ratios among phospholipid subclasses in 3 NAFLD models: Ob/Ob mice, HFD-, and MCD-fed mice. Ob/Ob mice, HFD-, and MCD-fed mice are described in Figure 1, Figure 3 and Figure 5, respectively. The ratio among PLs in livers of (**A**) Ob/Ob, (**B**) HFD-fed mice, and (**C**) MCD-fed mice. Data are mean ± SEM, N = 5–7 for (**A**); N = 4–5 for (**B**), and N = 5–6 for (**C**). #, *p* < 0.05, *versus* WT; *, *p* < 0.05, Ob/Ob *versus* Ob/Ob-iPLA_2_β KO or WT/HFD *versus* iPLA_2_β KO/HFD.

**Figure 8 biomolecules-10-01332-f008:**
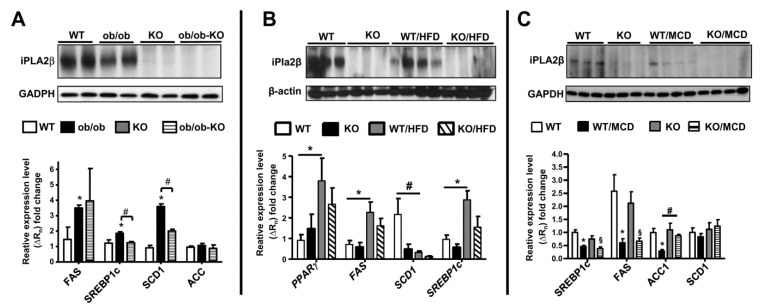
Expression of iPLA_2_β protein and de novo lipogenesis mRNA in livers of WT, Ob/Ob mice, HFD-, and MCD-fed mice. Ob/Ob mice, HFD-, and MCD-fed mice are described in Figure 1, Figure 3 and Figure 5, respectively. Expression of (**A**) iPLA_2_β protein, (**B**) HFD-fed mice, and (**C**) MCD-fed mice. Data are mean ± SEM, N = 5–7 for PCR data; #, *p* < 0.05; §, *p* < 0.05, KO *versus* KO/MCD; *, *p* < 0.05 between indicated groups.

**Figure 9 biomolecules-10-01332-f009:**
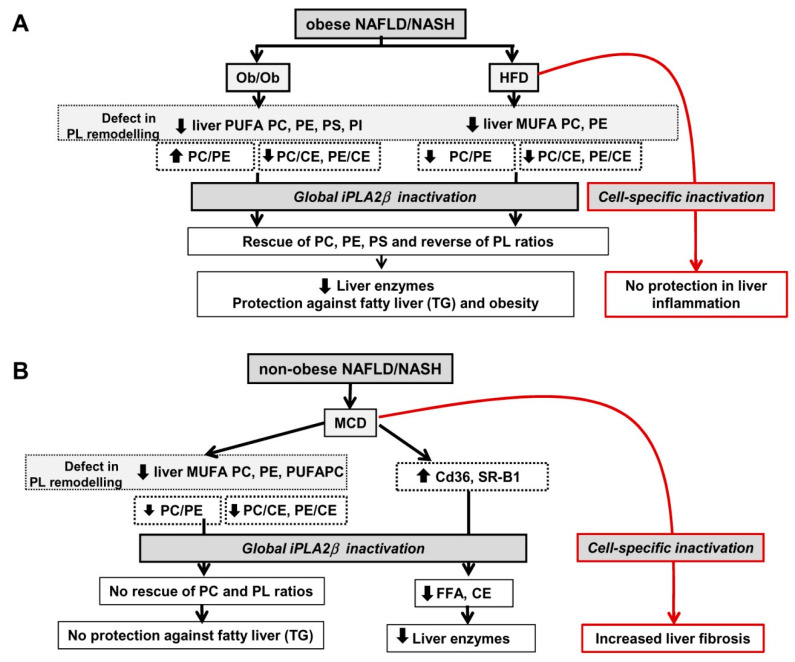
Role of iPLA_2_β deficiency in obese and non-obese NAFLD/NASH mouse models. (**A**) Livers of genetic Ob/Ob and chronic HFD-fed mice exhibited a defect in PL remodeling with suppressed contents of PUFA PLs. PC/PE ratio was increased in Ob/Ob mice while that of HFD-fed mice was decreased. iPLA_2_β inactivation replenished PLs associated with fatty liver protection. (**B**) Livers of MCD-fed mice exhibited a defect in PL remodeling with suppressed PUFA PLs as well as PC/PE ratio. iPLA_2_β inactivation in MCD-fed mice did not rescue this defect with no protection. We propose that iPLA_2_β deficiency in specific cell types may lead to no protection in liver inflammation and liver fibrosis in HFD and MCD NAFLD model, respectively (marked in red).

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
