# Peer review of "Rescue of Hepatic Phospholipid Remodeling Defect in iPLA2β-Null Mice Attenuates Obese but Not Non-Obese Fatty Liver"

_biomolecules, 2020, doi:10.3390/biom10091332_

Round 1
Reviewer 1 Report
The authors present a thorough review of iPLA2b involvement in rodent obese models. To balance the perspective, the authors should:
- Discuss the roles of iPLA2gamma, cPLA2alpha, and sPLA2s in obesity.
- Relate the lipid landscape in the rodent models with human obesity lipidome.
- Elaborate on the feasibility of using chemical inhibitors as therapeutics.
Also, can the authors present iPLA2b expression in the 3 models they studied and relate this to expression in human obesity (form various data bases).
Please be consistent with “iPLA2b” listing – “PLA” is capitalized and “2” is a subscript.
Author Response
Response to Reviewer#1
- We have included the discussion pertaining to other PLA2 enzymes, as shown on pages 3-4, lines 141-168.
- We have added references related to changes in phospholipids and PC/PE ratios in livers of human NAFLD and NASH, as shown on page 3, line 125-136, page 13, lines 465-478.
- The use of iPLA2β inhibitors is discussed under 13. Perspectives, section 13.2. The feasibility and consideration for tissue-specific iPLA2β inhibitors are discussed on page 16, lines 596-617.
- We have included a new Figure 8 showing iPLA2β protein expression in livers of our mice (page 12, lines 434-458). Unfortunately, no data have been published regarding iPLA2β protein in livers of human NAFLD and NASH.
- We have been following the nomenclature rules for gene and protein names among different species.
Gene names should be italicized; protein products of the loci are not italicized. For murine models, the gene and protein names are lowercase except for the first letter (e.g. gene: Abcb4; protein: Abcb4). For humans, the whole gene name is capitalized (e.g. gene: ABCB4; protein: ABCB4).
However, we did a global change from iPla2β to iPLA2β as suggested by the reviewer.
Reviewer 2 Report
This body of work submitted by the authors makes a significant contribution to the role of NAFLD and inflammation in obese and non-obese. Additionally, the authors' exploration of the role of iPla2β in liver macrophage inflammatory response and fibrosis adds to the growing knowledge of liver related morbidity.
Author Response
Thank you for your positive comments.
Reviewer 3 Report
Summary: This is a review of findings with iPLA2b-null mice in mouse models of fatty liver disorders prepared by leading and well-regarded investigators in this area. The authors observe that human obesity is achieving epidemic prevalence and that obesity-associated co-morbidities include Type 2 diabetes mellitus (T2DM), atherosclerotic cardiovascular disease (ASCVD), non-alcoholic fatty liver disease (NALFD), and non-alcoholic steatohepatitis (NASH). The authors have studied 3 mouse models of NALFD/NASH on the C57BL/6 genetic background, including leptin-deficient Ob/Ob mice, mice subjected to long term consumption of a high fat diet (HFD), and mice fed a methionine-choline deficient (MCD) diet. The first two models are if obese NAFLD and the third a model of non-obese NAFLD. Changes in hepatic phospholipids (PL) were characterized because PL are important surface components of lipid droplets; PL metabolism is involved in regulation of de novo lipogenesis by sterol-regulatory element binding protein (SREBP) transcription factors; and PL are involved in metabolism and secretion of very low density lipoproteins (VLDL). Humans with NAFLD/NASH exhibit a decrease in hepatic phosphatidylcholine (PC) and phosphatidylethanolamine (PE) content and a decline in PC species containing arachidonate (20:4) and in Triacylglycerol (TAG) species containing docosahexaenoate (22:6). The authors crossed Ob/Ob mice with global iPLA2b-null mice to produce Ob/Ob-iPLA2b-null mice, which were partially protected from the fatty liver phenotype compared to Ob/Ob mice. ESI/MS/MS analyses indicated that hepatic monounsaturated fatty acid (MUFA)-PL species declined in Ob/OB mice, as did polyunsaturated fatty acid (PUFA)-PL species, including PUFA-PC, PUFA-PE, and PUFA-PS species. These changes were blunted or absent on the livers of Ob/Ob-iPLA2b-null mice, suggesting that iPLA2b has a pathophysiologic role in depleting MUFA-PL and PUFA-PL species in the livers of Ob/OB mice. Liver endoplasmic reticulum (ER) fractions of HFD-fed mice exhibited reduced PC content, and iPLA2b deficiency protected against this effect, and similar protection was observed on the effect of HFD-feeding to reduce ER contnets of E and PS. In contrast, iPLA2b-deficiency magnified the changes in hepatic ER PL composition induced by MCD-dietary feeding.
Comments:
- The authors point to GWAS studies of obesity in humans and mice, but discussion of published data on the association between obesity, NASH, or NAFLD with iPLA2b genetic variants observed in such studies could be amplified for clarity. These issues are touched upon in the discussion of reference 57 at the top of page 4 of the review, and it might be helpful to expand this discussion.
- As the authors state on page 4, iPLA2b is a member of the PNPLA family, designated PNPLA9. This is of interest in view of the involvement of other members of this family, including PNPLA3, in metabolic disorders and further discussion of these issues might be helpful.
- Barbour and Ramanadham have reported of inter-relationships between iPLA2b and the SREBP pathway that are relevant to the subjects reviewed here, and some discussion of their obsevations might be helpful: Lei X, Zhang S, Barbour SE, Bohrer A, Ford EL, Koizumi A, Papa FR, Ramanadham S. Spontaneous development of endoplasmic reticulum stress that can lead to diabetes mellitus is associated with higher calcium-independent phospholipase A2 expression: a role for regulation by SREBP-1. J. Biol. Chem. 2010; 285 (9): 6693-705. PMID: 20032468.
- The authors describe several publications involving use of global iPLA2b-null mice to explore the role of iPLA2b in obesity and NALFD/NASH, but it is not clear whether they generated their own iPLA2b-null mice or used one of the previously reported lines, which include the lines described in Reference [69], the line in reference [70], and at least one other line, such as that described in the reference: Shinzawa K, Sumi H, Ikawa M, Matsuoka Y, Okabe M, Sakoda S, Tsujimoto Y. Neuroaxonal dystrophy caused by group VIA phospholipase A2 deficiency in mice: a model of human neurodegenerative disease. J. Neurosci 2008; 28 (9): 2212-20. It would be helpful if the authors could specify the nature and source of the global iPLA2b-null mice used in the primary studies that are reviewed in the current manuscript, especially if they do not correspond to one of the previously described lines.
- The authors mention the proposal several times in this manuscript that conditional iPLA2b-null mice that are selectively deficient in iPLA2b in only specific cells, such as hepatocytes or macrophages, would be a useful complement to the global iPLA2b-null mice for examining the involvement of iPLA2b in specific cellular processes. Is this based on experience with such conditional knockout mice? Have the authors used, for example Cre-Lox technology to develop such conditional knockouts or obtained such mice from others?
- The authors note in the fourth paragraph in section five on page 4 that "Despite PLA2G6 implications in obesity and NAFLD in mouse and human GWAS [57-60], HFD feeding of C57/BL6 mice for 6 months did not improve, but rather impaired glucose intolerance due to an impairment of insulin secretion by pancreatic islets in global deficient-iPLA2b KO mice [69]." I think the more appropriate citation here would be reference [61], which deals with glucose homeostasis in HFD-fed iPLA2b KO mice. Reference [69] deals instead with male infertility in iPLA2b KO mice.
Author Response
Response to reviewer#3
- Thank you for this comment. There was a mistake in referencing. In fact, ref. 57 (now 77) does not show association of NAFLD with PLA2G6 but with lysophospholipase-like 1. However, PLA2G6 is associated with body fat percentage (adiposity), blood TG, and Type-2 diabetes. We have added references with detailed discussion regarding GWAS association with iPLA2β, as shown on pages 4-5, lines 194-210.
- We have added discussion on PNPLA3 role in NAFLD in mice and humans , as shown on page 4, lines 165-167 and lines 181-192.
- Thank you for this reference that iPLA2b transcription is regulated by SREBP. We have added the new Figure 8 to include STRBP1c mRNA expression in livers of our mice. This reference and the disccussion now appear on pages 12, lines 433-457.
- In 2011, we obtained iPla2β-null mice from Dr. John Turk and we have been using this line in all of our studies. We have added this clarification on page 5, line 217 and in Acknowledgements on page 16, lines 626-627.
- Thank you for your comments. We have recently generated Pla2G6flox mice using Cre-Lox system for conditional knockouts. We are now phenotyping macrophage(LysM-Cre)-specific Pla2G6 deletion under various stress conditions.
- Thank you for this comment, and we have made this correction. Dr. Turk’s work was very extensive that we could not keep track of. We thank you again.
